# Quality of life disparities among Mexican people with systemic lupus erythematosus

Ana Laura Hernández-Ledesma[1]*, Domingo Martínez[1], Elizabeth Fajardo-Brigido[2], Talía V. Román-López[3], Karen J. Nuñez-Reza[1], Andrea Y. Tapia-Atilano[4], Sandra V. Vera del Valle[5], Donají Domínguez-Zúñiga[6], Lizbet Tinajero-Nieto[7], Angélica Peña-Ayala[7,8], Estefania Torres-Valdez[9], Gabriel Frontana-Vázquez[9], María Gutiérrez-Arcelus[10,11,12], Florencia Rosetti[13], Sarael Alcauter[3], Miguel E. Rentería[14,15], Alejandra E. Ruiz-Contreras[4], Deshiré Alpízar-Rodríguez[16], Alejandra Medina-Rivera[1]*

**1** Laboratorio Internacional de Investigación sobre el Genoma Humano, Universidad Nacional Autónoma de México, Campus Juriquilla, Blvd Juriquilla 3001, 76230 Santiago de Querétaro, México, **2** Instituto de Biotecnología, Universidad Nacional Autónoma de México, Cuernavaca, Morelos, México, **3** Departamento de Neurobiología Conductual y Cognitiva, Instituto de Neurobiología, Universidad Nacional Autónoma de México, Juriquilla, Querétaro, México, **4** Laboratorio de Neurogenómica Cognitiva, Unidad de Psicobiología y Neurociencias, Coordinación de Psicobiología y Neurociencias. Facultad de Psicología, Universidad Nacional Autónoma de México, Coyoacán, Ciudad de México, México, **5** Facultad de Medicina, Universidad Autónoma de Durango, Querétaro, México, **6** Private practice, Cuernavaca, Morelos, México, **7** Hospital General Regional No. 1, Instituto Mexicano del Seguro Social, Querétaro, México, **8** Instituto Nacional de Rehabilitación "Luis Guillermo Ibarra Ibarra", Ciudad de México, México, **9** Hospital General Regional No. 2, Instituto Mexicano del Seguro Social, Querétaro, México, **10** Division of Immunology, Boston Children's Hospital, Boston, Massachusetts, United States of America, **11** Harvard Medical School, Boston, Massachusetts, United States of America, **12** Program in Medical and Population Genetics, Broad Institute of Harvard and MIT, Cambridge, Massachusetts, United States of America, **13** Departamento de Inmunología y Reumatología, Instituto Nacional de Ciencias Médicas y Nutrición Salvador Zubirán, Tlalpan, Ciudad de México, México, **14** Mental Health & Neuroscience Program, QIMR Berghofer Medical Research Institute, Brisbane, QLD, Australia, **15** School of Biomedical Sciences, Faculty of Medicine, The University of Queensland, Brisbane, QLD, Australia, **16** Centro Médico Althea, Morelos, México

* alhernandezledesma@liigh.unam.mx (ALHL); amedina@liigh.unam.mx (AMR)

## Abstract

Higher prevalence and worst outcome have been reported among people with systemic lupus erythematosus with non-European ancestries, with both genetic and socioeconomic variables as contributing factors. In Mexico, studies assessing the inequities related to quality of life for Systemic Lupus Erythematosus patients remain sparse. This study aims to assess the inequities related to quality of life in a cohort of Mexican people with SLE. This study included 942 individuals with SLE from the Mexican Lupus Registry (LupusRGMX) and two healthy control groups. Self-answered surveys were collected via the Research Electronic Data Capture platform between May 2021 and January 2023. Data was analyzed as a cross-sectional study. A random forest model was implemented to assess potential predictive variables. Permutation tests were performed to analyze the effect health providers had on diagnosis lag and quality of life's differences among socioeconomic levels. Partial correlation analysis between the number of patients and rheumatologists registered was also performed. Systemic Lupus Erythematosus participants had significantly lower quality of life than healthy people (p-values < 0.0001). Socioeconomic status, delay in diagnosis, and corticosteroid consumption were the factors that influenced QoL the most (RMSE = 9.53 with the importance variable validated); lower quality of life was associated with lower

**Data Availability Statement:** All data is available upon request through the MexOMICS Consortium (https://redcap.link/nqsxtj8n). All the analyses were developed and implemented in R 4.2.3 language, and are available on https://github.com/NeuroGenomicsMX/Lupus_RGMX_analysis.

**Funding:** This project was supported by CONACYT-FORDECYT-PRONACES grant no. [11311] and [6390]. A.M.R. was supported by Programa de Apoyo a Proyectos de Investigación e Innovación Tecnológica–Universidad Nacional Autónoma de México (PAPIIT-UNAM) grants no. IA203021 and IN218023 and by Chan Zuckerberg Initiative Ancestry Network (2021-240438). A.L.H. L. is a doctoral student from Programa de Doctorado en Ciencias Biomédicas, Universidad Nacional Autónoma de México (UNAM). She received a fellowship CVU/Becario (711015/ 790972) from Consejo Nacional de Humanidades Ciencia y Tecnología (CONAHCYT). D.M. is a postdoctoral researcher supported by Consejo Nacional de Humanidades Ciencia y Tecnología (CONAHCYT), Estancias Posdoctorales por Mexico Convocatoria 2023(1), CVU 371892. M.E.R. thanks support from the Rebecca L Cooper Medical Research Foundation through an AI & Val Rosenstrauss Fellowship (F20231230). The funders had no role in study design, data collection and analysis, decision to publish, or preparation of the manuscript.

**Competing interests:** D.A.R is a scientific advisor for GSK Mexico, not related to this work. The authors declare they do not have any competing nor conflict of interest in connection with this article.

socioeconomic status (p-value < 0.0001). Disparities in health services were reflected in longer diagnosis time among people with public health providers (p-value = 0.0419). A significant association between diagnosed patients and available rheumatologists by geographical state was observed (ρ = 0.4, p-value = 0.0259), which can be translated into restricted access to specialists. Since most of our cohort exhibited low socioeconomic status, it is important to consider them as a vulnerable population; this study settles the necessity to deepen the effects of the socioeconomic disparities, allowing to design public policies and strategies aimed to reduce Systemic Lupus Erythematosus disparities, therefore improving quality of life of Mexican people with Systemic Lupus Erythematosus.

## Author summary

Systemic Lupus Erythematosus is an autoimmune disease that has been reported to be more prevalent and severe among people with non-European ancestries; still, there is scarce information regarding how SLE affects these other populations. SLE development is the result of the convergence of genetic, environmental, and social factors; Mexico exhibits a wide heterogeneous population, not only due to our genetic diversity, but also due to the high contrast observed at socioeconomic level across the country. Here, we evaluated the disparities in quality of life among a cohort of Mexican people with SLE, and we observed lower quality of life among our cohort when compared with people without SLE. This lower quality of life was associated with lower socioeconomic status, longer diagnosis time and the consumption of corticosteroids. Our study provides a first insight into the effects that socioeconomic disparities may have on the quality of life on the Mexican individual with SLE, future works will deepen the consequences of these inequities on our population.

## 1. Introduction

Systemic Lupus Erythematosus (SLE) is an autoimmune disease that exhibits widely heterogeneous manifestations. Disparities have been reported among people with SLE, not only regarding the prevalence, but also in outcomes associated with their clinical and psychological status; higher prevalence, disease activity and mortality rates have been reported among Latin-American, admixed North-American, African, and Native-American populations [1–4]. These disparities cannot be attributable only to genetic differences; demographic and social determinants should also be considered in the evaluation of the person with SLE [5]. According to the Regional Human Development Report 2021 of The United Nations Development Program, Latin-America exhibits great inequalities in economy, policies, employment, health access and violence; the impact that these inequalities may have on people with SLE in Latin-American countries has been scarcely explored. The limited representation of minorities hinders the generalizability and accessibility of knowledge and, therefore, limits the scope of treatments and diagnosis methods [2–4,6,7].

For Latin American populations, there are two important research initiatives: the Grupo Latino Americano de Estudio del Lupus (GLADEL) and the Lupus in Minorities: Nature vs Nurture (LUMINA) [8–10]. GLADEL aims to improve understanding of the characteristics and needs of Latin-American people with SLE, with a longitudinal cohort that has evaluated potential biomarkers, risk factors, treatments, and clinical features, representing a landmark

project. The GLADEL and GLADEL 2.0 cohorts included 248 and 145 Mexican individuals, respectively, from 7 different centers [9,10]. LUMINA cohort is a valuable resource for understanding how ethnicity affects SLE prevalence; it recruited, from 1993 to 2009, 640 participants with SLE from African American, Hispanic, and European ancestries, with a significant participation of people from Texas, who were mainly of Mexican ancestry [8,11].

Both, GLADEL and LUMINA cohorts, have proven not just the relevance of ethnicity in SLE, but also the influence of other health care associated disparities such as socioeconomic status, access to health insurance and prescribed treatments or social support in the course of the disease. The LUMINA cohort recently reported, in a study that included Hispanic people from Texas and Puerto Rico, that although genetics contributes significantly to SLE onset and outcomes, socioeconomic factors earn relevance through the disease evolution [11].

With over 126.7 million people, Mexico represents the third most populated country in America; Mexicans are the result of the admixture between Mexico's widely diverse native ethnic people and the millions of immigrants that have established from abroad the world. This admixture provides a wide genetic diversity that changes even between members of the same geographic region [12]; this genetic diversity, along with economic, health accessibility, cultural and even political disparities may influence the outcome of a Mexican individual with an SLE diagnosis. In Mexico, SLE prevalence has been estimated as 20 cases per 100,000 inhabitants, with no epidemiological reports on its prevalence or incidence, nor any epidemiological surveillance system to identify specific characteristics and unmet needs. Here, we describe disparities in socioeconomic, demographic, and clinical determinants and their contribution to quality of life in a Mexican SLE cohort [13].

## 2. Material and methods

### 2.1. Ethics

This project was approved by the Ethics on Research Committee of the Institute of Neurobiology at Universidad Nacional Autónoma de México (UNAM). Participant information was anonymized and stored at the National Laboratory of Advanced Scientific Visualization at UNAM. Participants provided digital signed informed consent, and a copy of the privacy statement was given to them, in accordance with the Federal Law on the Protection of Personal Data Held by Private Parties.

### 2.2. Patient selection criteria and recruitment

Participants were invited and recruited on a voluntary basis; through social media campaigns, a website (https://lupusrgmx.liigh.unam.mx/) and through invitations to the most active lupus communities in Mexico, rheumatologists, and the Mexican College of Rheumatology.

Eligible participants must be over 18 y/o, and willing to provide written informed consent and complete the study questionnaires. SLE participants must have a confirmed diagnosis and fulfill $\geq 4$ American College of Rheumatology (ACR) criteria [14]. For this analysis, the diagnosis of SLE was further confirmed if patients reported the current use or history of use of corticosteroids and immunosuppressants. Recruitment for this analysis was held between May 2021 and January 2023. Disease activity, treatment, and comorbidities were not considered as exclusion criteria.

### 2.3. Database collection platform

A digital data collection platform was implemented using the Research Electronic Data Capture (REDCap) software, which is hosted at UNAM [15,16]. Participants fulfilled, via online,

four self-perception surveys, namely: quality of life (QoL), sociodemographic, clinical, and socioeconomic questionnaire.

The Quality of life (QoL) questionnaire implemented the Spanish version of the World Health Organization QoL Questionnaire (WHOQOL-BREF), which includes 26 items validated on a 5-point Likert scale; 24 from WHOQOL-100 questionnaire, and two additional items to evaluate overall QoL and general health perception. This questionnaire considers the overall QoL score as well as four domain scores, namely, environmental, physical-health, psychological and social-relations scores. Higher scores are indicative of a better QoL [17].

The sociodemographic questionnaire included age, sex, state of residence, employment status, higher academic degree, and living arrangements variables. For comparison purposes, we also retrieved, from the official website, the number of medical specialists in rheumatology enrolled at the Mexican College of Rheumatology in every geographical Mexican state.

The clinical questionnaire included items about symptoms, disease activity, autoantibodies and serology tests, comorbidities, and treatment information. Due to the educational and socioeconomic heterogeneity of our cohort, we designed infographics that explained, in nontechnical language, each manifestation.

Lastly, the socioeconomic questionnaire assessed participant's socioeconomic status using the rule (NSE 2022) of the Mexican Association of Market and Opinion Intelligence Agencies (AMAI); an algorithm used to classify Mexican homes based on household welfare according to: human capital, practical infrastructure, connectivity and entertainment, health infrastructure, planning/future, and availability of basic infrastructure and space. It includes 6 items with a total of 300 points. It contemplates seven levels: A/B (202+ points), C+ (168–201), C (141–167), C- (116–140), D+ (95–115), D (48–94) and E (0–47). A higher score implies a better capacity to fulfill basic needs [18]. The high stratum comprises A/B and C+ levels, whereas C, C- and D+ levels represent the middle stratum, and levels D and E integrate the low socioeconomic stratum [19].

**2.3.4. Statistical analysis.** All curated, preprocessing, and data analyses were developed and implemented in R 4.3.1 [20]; the code is available at https://github.com/NeuroGenomicsMX/Lupus_RGMX_analysis. An automated script was implemented to find duplicates based on name, cell phone, and email. Completeness among duplicates was compared, and the most complete registry was kept. The script ran blind to maintain anonymity. To avoid false self-registries, following rheumatologists' suggestions, we established a filter for classical treatments used in SLE, corticosteroids and immunosuppressants; every participant consuming at least one SLE treatment was included in further analysis.

First, permutation tests were performed to contrast the overall QoL score, as well as environmental, physical-health, psychological, and social-relations scores between individuals with SLE (n = 1214, 1138 women, mean-age = 37.06, SD = 38.48) and two groups of healthy controls. The first control group (n = 179, 132 women, mean-age = 30.56, SD = 11.55) self-answered the questionnaires via the LupusRGMX platform. Whereas the second control group (n = 1271, 924 women, mean age = 29.29, SD = 49.04) self-answered questionnaires via the TwinsMX platform, TwinsMX is a registry established by the MexOMICS Consortium as well as LupusRGMX and therefore shares design and surveys [21]; for this analysis, only one individual per family was considered. Both control groups were recruited on a voluntary basis through social media campaigns. They must be at least 18 years old, be willing to provide written informed consent, and complete the questionnaires online.

After removing uncompleted registries, 1578 were kept: 942 individuals with SLE (888 women), 128 healthy individuals from the first control group (87 women), and 508 healthy individuals from the second control group (395 women). We corrected the imbalance between

groups by applying a random resample technique [22,23] with 200 resamples of n = 100, balancing 90% women and 10% men.

Second, a random forest model was applied, setting QoL as the outcome variable and age, sex, years living with SLE, treatment, comorbidities, health provider, diagnostic, diagnostic lag, and socioeconomic status as the predictive variables. After removing uncompleted and unmatched registries, we kept 712 cases. We split 70% of the data to train the model and left 30% for validation. Since we have noisy variables, we trained the base model running a search grid with the following hyperparameters: number of trees = 10 times number of predictive variables; sampling with and without replacement; resampling fraction: {0.4, 0.5, 0.6, 0.7, 0.8}; minimum node size: {1, 3, 5, 10, 20, 30, 40, 50}; mtry: {0.05, 0.15, 0.25, 0.33, 0.4, 0.6, 0.7, 0.8, 1} times number of predictive variables. All predictive variables were transformed to binary by one hot technique. Once we got the best parameters, we re-ran the model with such optimal parameters and 5000 trees to calculate the variables' importance using the permutation approach. Ultimately, we validated the variables' importance on the validation data partition, and marginal effects were calculated for the most influential variables [24].

Third, we explored the SLE individual's distribution among socioeconomic levels, then we compared the overall QoL scores among socioeconomic levels. After removing uncompleted registries, we kept 966 (906 women), on which we performed a pairwise permutation test [25] corrected by false discovery rate (FDR).

Fourth, based on the k-mean algorithm, we grouped participants by their delay in diagnosis time, which resulted in two groups. The first group had 506 patients; 339 of them were attending private health providers, whereas 167 were attending public health providers. The second group was composed of 160 patients; 106 of them were attending private health providers, and 54 of them were attending public health providers. Then we compared, in each group, private health providers' diagnostic times vs. public health providers' diagnostic times. For that purpose, we also ran permutation tests.

Finally, we ran a partial correlation analysis between the number of patients and rheumatologists registered in each geographic Mexican state while controlling the state's population size. In other words, we evaluate the degree of association between the number of patients and the number of rheumatologists in each state while removing the effect of population size in each state. We wanted to understand how the number of SLE diagnostics is related to the number of available medical specialists in rheumatology.

Note that, all tests and models applied in the present study are non-parametric, and they can handle continuous and categorical data [24–33]. However, given that our questionnaire design presented the Likert scale equally spaced, showing visual symmetry around the central point, and kept precisely the same range for all the items, we also would be able to treat and interpret them as numeric variables into the interval scale [34,35]. Moreover, aggregated questionnaires (e.g, WHOQoL and NSE 2022) could be handled as continuous because of the central limit theorem [34,36].

## 3. Results

Regarding QoL, we found significant differences between the SLE-group and both healthy control groups for the overall score and all domains of QoL (see Fig 1). In all cases, the SLE-group scored the lowest. For the overall score (Fig 1A), the SLE-group scored 7.67 (SE = 1.26) points lower than Control group 1 (p-value < 0.0001); and 10.49 (SE = 1.21) points lower than Control group 2 (p-value < 0.0001).

For the environmental domain (Fig 1B), SLE-group scored, on average, 1.12 (SE = 0.33, p-value = 0.0111) and 2.10 (SE = 0.32, p-value<0.0001) lower than Control group 1 and Control

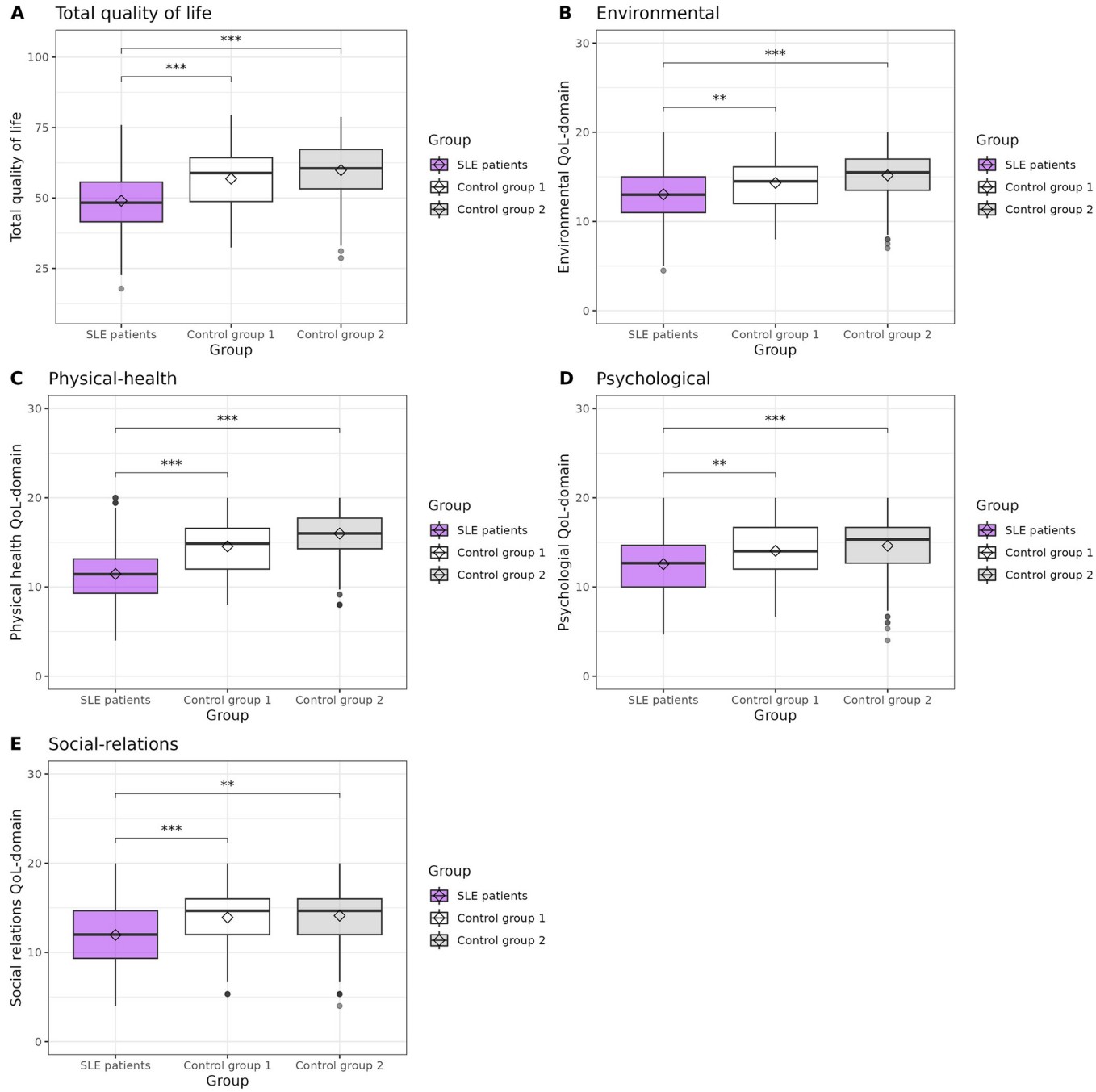

**Fig 1.** Quality of life scores comparisons by permutation test among SLE-group (n = 942) and healthy controls individuals (Control group 1, n = 128; Control group 2, n = 395). Significant differences between SLE patients and both control groups were observed in total score (panel A) and all domains of QoL (panels B, C, D and E). The SLE-group always scored the lowest.

group 2, respectively. Regarding physical-health domain (Fig 1C), SLE-group scored 2.89 (SE = 0.32, p-value<0.0001) and 4.30 (SE = 0.33, p-value<0.0001) points below Control group 1 and Control group 2, respectively. Regarding psychological domain (Fig 1D), SLE individuals rated 1.61 (SE = 0.39, p-value = 0.0015) and 2.18 (SE = 0.41, p-value<0.0001) points lower than Control group 1 and Control group 2 volunteers. For social-relations (Fig 1E), SLE-group

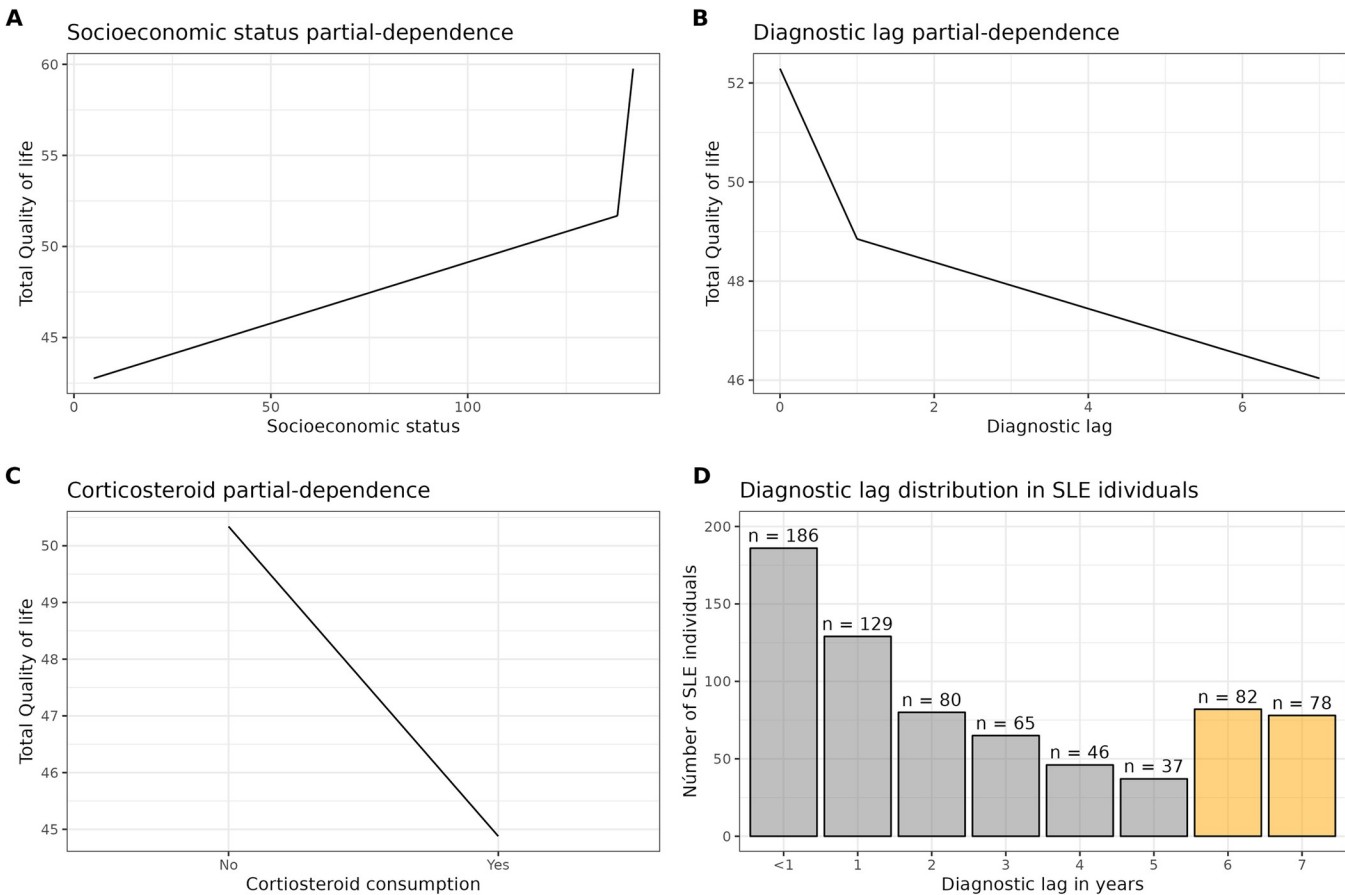

**Fig 2. Variable contribution to quality of life (n = 942).** Effect of socioeconomic status (A), diagnostic lag (B), and corticosteroid consumption (C) over total QoL. Higher socioeconomic status is associated with better QoL, whereas greater delay in diagnosis (i.e., diagnostic lag) and corticosteroid consumption are associated with lower QoL. Number of SLE patients and their diagnostic lag in years (D). Yellow bars represent a cluster where an atypical number of patients got a very delayed SLE diagnostic, the rest of the bars (gray bars) constitute a group where the number of SLE patients decays exponentially as the diagnostic lag increases.

punctuated 2.13 (SE = 0.47, p-value = 0.0006) and 2.01 (SE = 0.48, p-value = 0.0040) below Control group 1 and Control group 2 individuals, respectively.

Concerning the random forest model, it showed a RMSE = 9.5 in the training dataset (70% of data). We found that socioeconomic status, delay in diagnostic (or diagnostic lag), and corticosteroid consumption are the variables with the greater impacts on QoL. We used the permutation-based approach to estimate such importance. The validation dataset confirmed the importance (RMSE = 9.8, 30% of the data); in addition, we ran a multivariate adaptive regression splines model, and the variable importance was corroborated, but in distinct order (see S1 Fig) In other words, socioeconomic status, diagnostic lag, and corticosteroid consumption, are the most important variables that consistently come out. The effect of these three variables over QoL can be seen in Fig 2, which is shown through their partial dependence plots. Higher socioeconomic status is associated with better QoL (Fig 2A), whereas greater diagnostic lag (Fig 2B) and corticosteroid consumption (Fig 2C) are associated with lower QoL.

Referring to the delay in diagnosis, panel D of Fig 2 shows such diagnosis time distribution, i.e., the number of SLE patients and their diagnostic lag grouped by year. The k-mean algorithm integrated patients with atypically longer diagnosis time, greater than 5 years, into one clustering (see yellow bars). For the rest of the bars (see gray bars), the number of SLE patients

decays exponentially as the diagnostic lag increases up to 5 years. For that group, where the number of SLE patients decays exponentially as the diagnostic lag increases (gray bars group), we found that public health providers take 0.3 years longer to diagnose SLE than private health providers (p-value = 0.0419).

Pertaining to the socioeconomic status, SLE-population distribution among each stratum can be seen in panel A of Fig 3, n = 1172. It is worth noting that most SLE individuals belong to the low and down-middle stratum. Box plots in panel B show population distribution into each stratum. We found statistical differences for QoL among no contiguous strata (adjusted p-values < 0.001, n = 966), which confirmed the partial (marginal) effect of socioeconomic stratum over QoL, shown in Fig 2 (A).

Finally, we evaluated the shortage of rheumatologists by geographical state. We found a significant association between the number of rheumatologists registered by each state and the SLE diagnosed individuals in each state ($\rho$ = 0.4, p-value = 0.0259) after removing the effect of population size (S2 Fig).

## 4. Discussion

Previous studies have reported disparities among prevalence and outcomes in people with SLE, especially among people with African and Hispanic ancestries. Both genetic and social factors are proposed to be contributing to SLE development [5] and are needed to be studied in susceptible populations to deepen the impact that these inequalities may have on people with SLE populations. This study evaluated disparities in socioeconomic, demographic, and clinical determinants and their contribution to quality of life in a Mexican SLE cohort.

Quality of life (QoL) is a concept that considers the self-perception that individuals have regarding their physical, psychological, social, and environmental conditions. Due to its potential to reflect the individual status, in late years, it has been proposed that self-perceived QoL should be considered as an important outcome in the design and decision-making of SLE treatment/policies. The LupusRGMX cohort exhibited a significantly lower total QoL score than control groups (Fig 1A); which has been previously reported in other populations and has been associated with other variables such as disease burden, pain, brain fog, anxiety, and fatigue [37,38]. This pattern was observed also when evaluating environmental (Fig 1B), physical health (Fig 1C), psychological (Fig 1D) and social relations (Fig 1E) domains. This affection in all domains may be related to daily life aspects such as retaining employment, access to health insurance or treatments affordability, which can enhance emotional stress, triggering mental health manifestations such as fatigue, anxiety, and depression, all associated with a diminished QoL [39,40]. The most significant difference between SLE group and Control groups was in the physical domain, in which the SLE group scored 2.89 and 4.30 points under the Control 1 and Control 2 groups, respectively. This result can be attributable to the manifestations that SLE exerts: diminished strength in extremities, joint swelling and pain, impaired physical function along with fatigue; which implies important physical limitations for the person [41].

To evaluate which variables may be influencing QoL, a random forest model was applied. Socioeconomic status, delay in diagnosis and corticosteroid consumption were the variables with greater influence on QoL (Fig 2). Higher socioeconomic status is associated with better QoL (Fig 2A), which is in line with previous studies that have proposed that health care system structure may not be equally accessible for all; better quality of care service is reported among specialized clinics, which are often centralized in big cities and have higher cost, whereas public health care system usually have great demand, which translates in less time available per patient with the specialist [5].

**A**

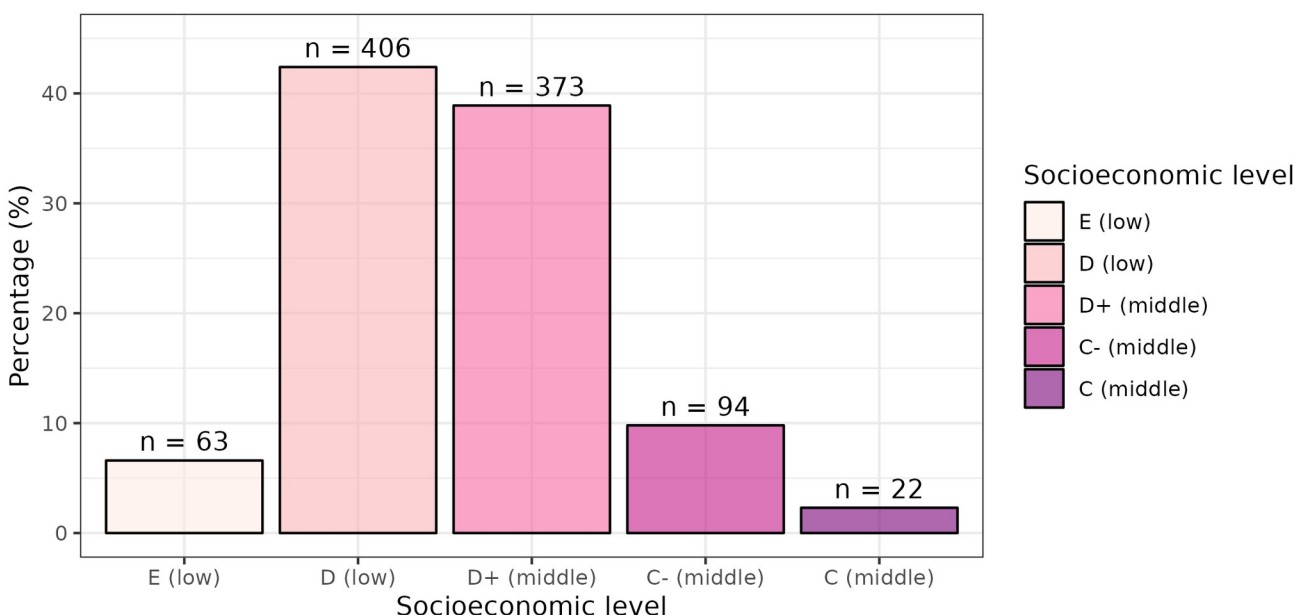

**B**

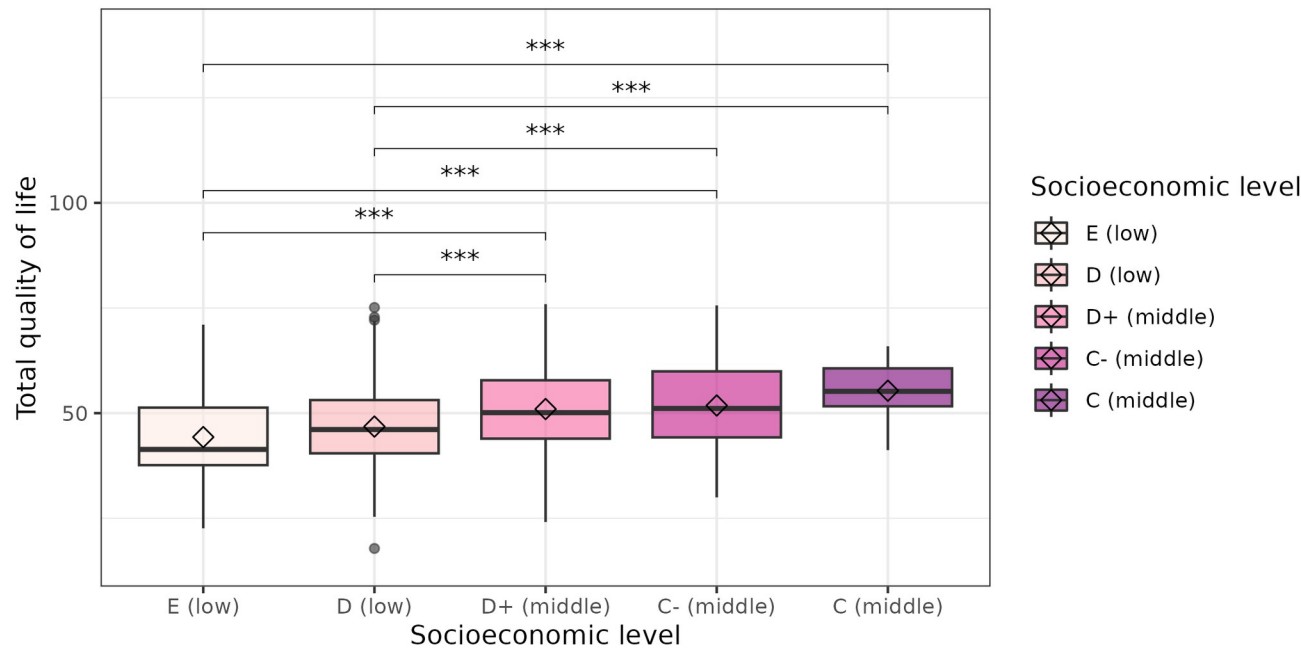

**Fig 3. Socioeconomic status and QoL of SLE patients.** Distribution of participants according to socioeconomic stratum (A); most belong to low and down-middle stratum. QoL comparison among socioeconomic levels (B) by permutation test corrected by FDR, higher socioeconomic levels show better QoL (n = 942).

In addition, SLE is a disease with high health care-related costs with an estimated cost of US $3000–12000 per year, in Mexico this estimation does not exist [42]. In contrast, larger delay in diagnosis is associated with worse QoL (Fig 2B); to support this, it has been reported that earlier diagnosis is associated with timely initiation of treatment, decreased risk of flares, lower tissue damage and better outcome. An improvement in the prognosis is also associated with a lower use of healthcare resources and infrastructures, implying lower healthcare costs [43].

Worse QoL was observed in individuals with consumption of corticosteroids (Fig 2C); this can be due to glucocorticoids, although effective, they have been related to neuropsychiatric manifestations; additionally, side effects include weight gain and other appearance-related effects, which can affect self-perception [44,45]. The sum of these factors, in addition to physical limiting manifestations, such as arthritis, influences health and QoL self-perception, explaining why these are the main variables contributing to QoL in our cohort. Since corticosteroids are cheaper than immunosuppressants, more research is needed to see if patients from lower socioeconomic levels are more likely to be treated with steroids. And, if patients from higher socioeconomic levels are more likely to be treated with immunosuppressants.

According to the NSE 2022, our cohort belongs to the lower five socioeconomic levels, people within these levels use 37–52% of incomes for food, and the rest to cover other necessities including healthcare [17] (Fig 3A). In Mexico, lower socioeconomic level is related to fewer possibilities to access healthcare, which translates into challenges to accessing medical specialists, longer waiting times for evaluations, difficulties in obtaining an early and accurate diagnosis, shorter duration of appointments (due to saturation) and low availability of prescribed drugs, which are also expensive [46]. Deepening into these variables, we observed significant differences in WHOQoL score among socioeconomic levels (Fig 3B).

Until January 2023, the LupusRGMX platform collected 1214 registries, from the 32 states of Mexico. Since some states exhibited lower registries, we looked up the rheumatologists enrolled at the Mexican College of Rheumatology and observed the similar geographic distribution as the participants' (**S2 Fig**). In Mexico, it is calculated that there is 0.55 certified rheumatologist per 100,000 inhabitants, with a surfeit in the capital and metropolitan areas [46,47]. This shortage can lead to unequal access to health and medical services [48]; government strategies are needed to improve health-care availability for people with SLE.

Limitations of this study include low participation in some areas, accessibility to the platform, the self-reported nature of surveys, lack of representation of some socioeconomic status and standard measures to assess SLE. One of our concerns was the lack of internet access in lower socioeconomic status, however, the prevalence of individuals among the lowest socioeconomic levels gives us a preliminary idea of how this population is being affected. As for the self-reported nature of surveys, recent work highlights the importance of considering patient's perception, as well as mental health related manifestations, in the line of people with SLE management, as this can bring under-detected needs to the conversation and have a positive impact on other aspects such as adherence to treatment; further analyses will focus on the validation of the self-application questionnaires and to the integration with standard measurements for SLE assessed by rheumatologist, such as disease activity or damage [46,47,49]. The sum of these limitations represents an opportunity to improve data quality and to enrich its analysis.

This work settles a first step into the understanding of the quality of life in Mexican people with SLE and the potential variables contributing to it. As previously mentioned, quality of life evaluation represents a valuable tool to assess the effect that a chronic disease such as SLE, have in the life of the patient. The interdisciplinary and synergistic work between clinicians, researchers, and people with SLE through the implementation of the LupusRGMX platform, has established a pioneering approach that provides a valuable source of data about QoL in

Mexican people with SLE. The information recovered and the understanding of the contribution of socioeconomic level on QoL will provide the data that allow us to identify the necessities of our population and that support the design of better strategies for diagnosis and follow up, and public policies that fulfill those necessities, and therefore will help to improve QoL in Mexican individuals with SLE.

## 5. Conclusion

Our data showed that Mexican people with SLE had significantly lower QoL than healthy people. Disparities in socioeconomic status exerts a significant influence on QoL in our cohort; lower incomes can be translated into less accessibility to specialists, which usually are centralized in the biggest cities; in addition, the use of public health system has shown association with longer diagnosis times. Importantly, most lupus patients in Mexico have low or down-middle socioeconomic status, making them a vulnerable population. The realization of this study establishes a multidisciplinary approach into integrate social information in the management of the individual with SLE. Further studies are needed to explore other variables associated with socioeconomic status and with their potential relationship with clinical variables; this data will add to our understanding of how SLE affects Mexican population and will also provide the data that support the request for new strategies and policies aiming to improve and speed up the process of diagnose and management of SLE in Mexico, considering the most vulnerable populations, therefore improving the QoL of the people with SLE in our country.

## Supporting information

**S1 Fig. Importance of predictive variables in training dataset.** (A) Random forest model and (B) multivariate adaptative regression splines model.
(TIF)

**S2 Fig. Geographic distribution of patients and specialists by state.** (A) Number of patients by state, and (B) number of specialists (rheumatologists) by state. Map created in R, with source shape files from Instituto Nacional de Estadistica y Geografia (INEGI). (2022). Marco Geoestadistico Estados Unidos Mexicanos. Recovered from [https://www.inegi.org.mx/app/biblioteca/ficha.html?upc=889463770541]. Terms of use: [https://www.inegi.org.mx/inegi/terminos.html].
(TIF)

## Acknowledgments

This work received support from Luis Aguilar, Alejandro León, and Jair García of the Laboratorio Nacional de Visualización Científica Avanzada. We also thank Carina Uribe Díaz, and Alejandra Castillo Carbajal for their technical support. Authors would like to express their special acknowledgment to Fundación Proayuda Lupus Morelos A.C, Lupus MX, El despertar de la Mariposa, and Laura Athié, co-director of Centro de Producción de Lecturas, Escrituras y Memorias [LEM], for their invaluable support.

## Author Contributions

**Conceptualization:** Ana Laura Hernández-Ledesma, Domingo Martínez, María Gutiérrez-Arcelus, Florencia Rosetti, Sarael Alcauter, Miguel E. Rentería, Alejandra E. Ruiz-Contreras, Deshiré Alpízar-Rodríguez, Alejandra Medina-Rivera.

**Data curation:** Ana Laura Hernández-Ledesma, Domingo Martínez, Elizabeth Fajardo-Brigido, Talía V. Román-López, Karen J. Nuñez-Reza, Andrea Y. Tapia-Atilano, Alejandra E. Ruiz-Contreras, Deshiré Alpízar-Rodríguez.

**Formal analysis:** Ana Laura Hernández-Ledesma, Domingo Martínez, Alejandra E. Ruiz-Contreras, Deshiré Alpízar-Rodríguez, Alejandra Medina-Rivera.

**Funding acquisition:** Alejandra Medina-Rivera.

**Investigation:** Ana Laura Hernández-Ledesma, Domingo Martínez, Elizabeth Fajardo-Brigido, Talía V. Román-López, Karen J. Nuñez-Reza, Andrea Y. Tapia-Atilano, Sandra V. Vera del Valle, Donají Domínguez-Zúñiga, Lizbet Tinajero-Nieto, Angélica Peña-Ayala, Estefania Torres-Valdez, Gabriel Frontana-Vázquez, Deshiré Alpízar-Rodríguez.

**Methodology:** Domingo Martínez, Alejandra Medina-Rivera.

**Project administration:** Alejandra Medina-Rivera.

**Writing – original draft:** Ana Laura Hernández-Ledesma, Domingo Martínez.

**Writing – review & editing:** Ana Laura Hernández-Ledesma, María Gutiérrez-Arcelus, Florencia Rosetti, Sarael Alcauter, Miguel E. Rentería, Alejandra E. Ruiz-Contreras, Deshiré Alpízar-Rodríguez, Alejandra Medina-Rivera.

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
