## [Decision Letter · Decision Letter 0]

28 May 2024

PDIG-D-23-00314

Quality of life disparities among Mexican people with systemic lupus erythematosus.

PLOS Digital Health

Dear MSc Ana Laura Hernandez-Ledesma,

Thank you for submitting your manuscript to PLOS Digital Health. After careful consideration, we feel that it has merit but does not fully meet PLOS Digital Health's publication criteria as it currently stands. Therefore, we invite you to submit a revised version of the manuscript that addresses the points raised during the review process.

Please submit your revised manuscript within 60 days Jul 27 2024 11:59PM. If you will need more time than this to complete your revisions, please reply to this message or contact the journal office at digitalhealth@plos.org. Please include the following items when submitting your revised manuscript:

We look forward to receiving your revised manuscript.

Kind regards,

Cleva Villanueva, M.D., Ph.D.

Guest Editor

PLOS Digital Health

Journal Requirements:

1. Please amend your detailed online Financial Disclosure statement. This is published with the article. It must therefore be completed in full sentences and contain the exact wording you wish to be published.

a) State the initials, alongside each funding source, of each author to receive each grant. For example: "This work was supported by the National Institutes of Health (####### to AM; ###### to CJ) and the National Science Foundation (###### to AM)."

2. Please provide separate figure files in .tif or .eps format only and remove any figures embedded in your manuscript file. Please also ensure that all files are under our size limit of 10MB. You may leave the figure captions or legends in the manuscript.

For more information about how to convert your figure files please see our guidelines: https://journals.plos.org/digitalhealth/s/figures

3. We have noticed that you have cited supplementary material "Supplementary Figure 1" on page 21 in your manuscript. However, there is no corresponding file uploaded to the submission. Please upload it as separate file with the item type 'Supporting Information'. 

Please ensure that the Supporting Information file has a legend listed in the manuscript after the references list.

4. We noticed that you used "data not shown" in the manuscript. We do not allow these references, as the PLOS data access policy requires that all data be either published with the manuscript or made available in a publicly accessible database. Please amend the supplementary material to include the referenced data or remove the references.

Additional Editor Comments (if provided):

The review process for your manuscript is complete. Based on the reviewers' comments, I concluded to accept his manuscript with further revision.

It is important that in your review you take into account all the reviewers' comments, especially in the methodological part and comments on the limitations of the study.

Once you have reviewed your manuscript, please follow the instructions for submitting it

Reviewers' comments:

Reviewer's Responses to Questions

**Comments to the Author**

1. Does this manuscript meet PLOS Digital Health’s publication criteria? Is the manuscript technically sound, and do the data support the conclusions? The manuscript must describe methodologically and ethically rigorous research with conclusions that are appropriately drawn based on the data presented.

Reviewer #1: Yes

Reviewer #2: Yes

2. Has the statistical analysis been performed appropriately and rigorously?

Reviewer #1: Yes

Reviewer #2: I don't know

3. Have the authors made all data underlying the findings in their manuscript fully available (please refer to the Data Availability Statement at the start of the manuscript PDF file)?

Reviewer #1: Yes

Reviewer #2: Yes

4. Is the manuscript presented in an intelligible fashion and written in standard English?

Reviewer #1: Yes

Reviewer #2: Yes

5. Review Comments to the Author

Reviewer #1: In the introduction, add the citation regarding the prevalence of SLE. Where were the statistics obtained from?

In the method section, add a list of definitions for the key terms used in the study.

Regarding the parameters collected, add a table showing the main parameters and by how many participants were reported. Were there any missing values?

The RMSE should be reported for both the calibration and test sets. The closer both values the more robust is the model.

Comment on the generalisability of the findings.

Could do with a language revision and proof reading.

Reviewer #2: Dear Author,

Your manuscript on "Quality of life disparities among Mexican people with systemic lupus erythematosus" presents valuable insights into an important topic. However, several aspects of the manuscript need attention to enhance its quality and impact.

Firstly, there is a need for more detailed explanations regarding the choice of parameters in your models, particularly the hyperparameters in the random forest model. This will help readers understand the rationale behind your modeling decisions and ensure transparency and reproducibility.

Secondly, the justification for using non-parametric tests and treating Likert scale data as numeric variables should be provided. Clarifying these methodological choices will strengthen the validity of your statistical analysis.

Additionally, the literature review would benefit from including more recent studies focusing on the Mexican population or similar contexts. Also, consider broadening the range of studies from different geographical regions to provide a more comprehensive overview of the topic.

Methodologically, it's essential to include information on the validation of the questionnaires used in the Mexican context. Further elaboration on the rationale behind choosing specific statistical tests and models will improve the clarity and rigor of your methodology section.

In the discussion section, thoroughly discussing the limitations of the study, including potential biases in self-reported data and the representativeness of the sample, is crucial. Furthermore, providing specific recommendations for policy or clinical practice based on your findings and linking them more explicitly to the existing literature will enhance the relevance and impact of your study.

Overall, addressing these comments will strengthen your manuscript and ensure it makes a meaningful contribution to the field.

6. PLOS authors have the option to publish the peer review history of their article (what does this mean?). If published, this will include your full peer review and any attached files.

**Do you want your identity to be public for this peer review?** For information about this choice, including consent withdrawal, please see our Privacy Policy.

Reviewer #1: No

Reviewer #2: Yes: Ali Akbar

---

## [Decision Letter · Decision Letter 1]

22 Nov 2024

Quality of life disparities among Mexican people with systemic lupus erythematosus.

PDIG-D-23-00314R1

Dear Ana Laura Hernández-Ledesma,

We are pleased to inform you that your manuscript 'Quality of life disparities among Mexican people with systemic lupus erythematosus.' has been provisionally accepted for publication in PLOS Digital Health.

Best regards,

Cleva Villanueva, M.D., Ph.D.

Guest Editor

PLOS Digital Health

The authors have answered all the commentaries and made the changes accordingly. This guest editor decides that the manuscript is suitable for publication at PLOS Digital Health
